# Clinical Characteristics and Local Histopathological Modulators of Endometriosis and Its Progression

**DOI:** 10.3390/ijms25031789

**Published:** 2024-02-01

**Authors:** Anca-Maria Istrate-Ofiţeru, Carmen Aurelia Mogoantă, George-Lucian Zorilă, Gabriela-Camelia Roşu, Roxana Cristina Drăguşin, Elena-Iuliana-Anamaria Berbecaru, Marian Valentin Zorilă, Cristina Maria Comănescu, Stelian-Ștefăniță Mogoantă, Constantin-Cristian Vaduva, Elvira Brătilă, Dominic Gabriel Iliescu

**Affiliations:** 1Department of Histology, University of Medicine and Pharmacy of Craiova, 200349 Craiova, Romania; ancaofiteru92@yahoo.com (A.-M.I.-O.); nicola_camelia92@yahoo.com (G.-C.R.); 2Research Centre for Microscopic Morphology and Immunology, University of Medicine and Pharmacy of Craiova, 200349 Craiova, Romania; 3Department of Obstetrics and Gynecology, University Emergency County Hospital, 200642 Craiova, Romania; roxy_dimieru@yahoo.com (R.C.D.); dominic.iliescu@yahoo.com (D.G.I.); 4ENT Department, Faculty of Medicine, University of Medicine and Pharmacy of Craiova, 200349 Craiova, Romania; carmen_mogo@yahoo.com; 5Department of Obstetrics and Gynecology, University of Medicine and Pharmacy of Craiova, 200349 Craiova, Romania; hitmed@gmail.com; 6Doctoral School, University of Medicine and Pharmacy of Craiova, 200349 Craiova, Romania; 7Department of Forensic Medicine, University of Medicine and Pharmacy of Craiova, 200349 Craiova, Romania; zorilavaly@yahoo.com.au; 8Department of Anatomy, University of Medicine and Pharmacy of Craiova, 200349 Craiova, Romania; cristinacomanescu85@gmail.com; 9General Surgery Department, University of Medicine and Pharmacy of Craiova, 200349 Craiova, Romania; ssmogo@yahoo.com; 10Department of Obstetrics and Gynecology, “Carol Davila” University of Medicine and Pharmacy, 050474 Bucharest, Romania; elvirabarbulea@gmail.com

**Keywords:** endometriosis, adenomyosis, multiple immunohistochemistry

## Abstract

Endometriosis (E) and adenomyosis (A) are associated with a wide spectrum of symptoms and may present various histopathological transformations, such as the presence of hyperplasia, atypia, and malignant transformation occurring under the influence of local inflammatory, vascular and hormonal factors and by the alteration of tumor suppressor proteins and the inhibition of cell apoptosis, with an increased degree of lesion proliferation. Material and methods: This retrospective study included 243 patients from whom tissue with E/A or normal control uterine tissue was harvested and stained by histochemical and classical immunohistochemical staining. We assessed the symptomatology of the patients, the structure of the ectopic epithelium and the presence of neovascularization, hormone receptors, inflammatory cells and oncoproteins involved in lesion development. Atypical areas were analyzed using multiple immunolabeling techniques. Results: The cytokeratin (CK) CK7+/CK20− expression profile was present in E foci and differentiated them from digestive metastases. The neovascularization marker cluster of differentiation (CD) 34+ was increased, especially in areas with malignant transformation of E or A foci. T:CD3+ lymphocytes, B:CD20+ lymphocytes, CD68+ macrophages and tryptase+ mast cells were abundant, especially in cases associated with malignant transformation, being markers of the proinflammatory microenvironment. In addition, we found a significantly increased cell division index (Ki67+), with transformation and inactivation of tumor suppressor genes p53, B-cell lymphoma 2 (BCL-2) and Phosphatase and tensin homolog (PTEN) in areas with E/A-transformed malignancy. Conclusions: Proinflammatory/vascular/hormonal changes trigger E/A progression and the onset of cellular atypia and malignant transformation, exacerbating symptoms, especially local pain and vaginal bleeding. These triggers may represent future therapeutic targets.

## 1. Introduction

### 1.1. Clinical Features, Pathogenesis and Risk Factors

A represents a particular variety of endometrial ectopic localization where the endometrial tissue is located in the myometrial structure [1,2,3]. In A, the basal endometrium invades the hyperplastic myometrial fibers. In contrast to the functional layer, the basal layer does not typically undergo cyclic changes with the periodic cycle [4]. A can involve the uterus in a focal pattern, creating an adenomyoma. In the case of diffuse invasion, the uterus becomes enlarged and heavier [5]. The incidence of A was reported as 1.03% of all women of reproductive age [6]. E is a chronic gynecological disease characterized by the development and presence of histological elements such as endometrial glands and stroma in anatomical positions and organs outside the uterine cavity [7]. E is a benign condition often found in women of reproductive age. This pathology has an incidence of 10% of the general population of reproductive age. E is characterized by the migration of glands and endometrial stromal cells from the uterine mucosa at a greater distance to the fallopian tubes, ovaries, recto-vaginal space, uterine ligaments and peritoneum. It can also be found in other organs such as the urinary tract, lungs or in the abdominal wall after C-section delivery [8]. A can occur together with E; this varies in that patients with E have endometrial-like tissue located entirely outside the uterus. In the case of E, the endometrial tissue is similar to, but not the same as, the normal endometrium. The two conditions are found together in most cases, but often occur independently [9].

E and A have a major impact on women both physically and emotionally, affecting their quality of life through distressing symptoms such as severe dysmenorrhea, dyspareunia, chronic pelvic pain, metrorrhagia or menorrhagia and infertility [10].

The pathway leading to E is not fully understood, and multiple mechanisms were suggested to explain its pathogenesis. Retrograde menstruation, by which menstrual blood refluxes through the fallopian tubes and implants in pelvic structures, is frequently encountered and is associated with E; other mechanisms described are coelomic metaplasia, vascular or lymphatic spread with metastatic emboli and local immunological factors [11].

The risk factors associated with E are nulliparity, early onset of menstruation, menorrhagia (Men), late onset of menopause and local inflammatory factors. Multiparity, the use of oral contraceptives, breastfeeding and total hysterectomy with bilateral adnexectomy are protective factors [12].

### 1.2. Morphological Characteristics of Endometriosis and Adenomyosis

Macroscopically, E foci may present as superficial gunpowder burn lesions on the ovarian surface or the parietal or visceral peritoneum. The color of the lesions can range from white to red or shades of brown or blue. E lesions may also present as deep nodules or cysts containing blood loaded with macrophages that have phagocytosed hemosiderin—“chocolate cyst aspect”. The areas described may be surrounded by perilesional fibrosis of varying degrees. There are situations where E foci have atypical characteristics and appear as red implants (vesicular, polypoid, petechiae, hemorrhagic, flame-like) or transparent blisters [13].

E lesions are characterized by the presence of glandular epithelial tissue and peri-glandular stroma, and A is characterized by the presence of glandular epithelial structures implanted in the myometrium at a distance from the endometrial lining. Endometriotic lesions vary in glandular aspect, lesion size and location [13].

Although these are benign pathologies, they may also have some malignant characteristics such as invasive or metastatic potential. Malignant degeneration may occur under the influence of local inflammatory, vascular or hormonal factors that contribute to the development of atypical hyperplastic and subsequently malignant lesions. The malignant aspect must be carefully distinguished when there are deep infiltrating nodules or intra- cystic papillary projections [14]. E and A are not malignant disorders, but some characteristics are similar to cancer: they can develop local and distant foci, develop resistance to apoptosis and invade other tissues with subsequent damage [15]. These diseases generate a chronic local and systemic inflammatory environment [16] that can increase the risk of malignant transformation [8]. E has been associated with a higher risk of several cancers in population-based trials [15]. Gene sequencing has shown that approximately 20% of ovarian endometriosis (OE) and deep endometriosis lesions have cancerous somatic driver mutations [17]. Cancer arising with A is very rare, with transformation occurring in only 1% of cases [18,19,20]. The first case of clear cell adenocarcinoma (CACC) and endometrioid carcinoma (EC) resulting from A was reported in 1897 [21]. A is more frequently associated with EC, but CACC has also been noted [18,21].

Deep, nodular, infiltrative E extending > 5 mm into the retroperitoneum may affect the ligaments of the uterus (uterosacral, round), vagina, rectovaginal septum, bladder, ureters, bowel, appendix, colon or rectum. E structures may also take the form of mucous or serous polypoid masses that resemble a malignant lesion [13].

Histopathological examination is the mainstay for the final diagnosis of E or A and the extent of tissue proliferation. Classical histological staining, hematoxylin–eosin (HE), highlights endometrial glandular structures, the presence of peri-glandular stroma and the degree of atypical glandular transformation [14].

Immunohistochemical studies can highlight the progressive potential of E or A foci; the presence of an inflammatory microenvironment, neovascularization or hormonal receptors; or the involvement of tumor proteins in the preneoplastic or malignant transformation of E lesions [14].

### 1.3. Diagnosis and Treatment of Endometriosis and Adenomyosis

The diagnosis of A is suggested by clinical symptoms and is supported by transvaginal ultrasonography and pelvic magnetic resonance imaging (MRI). The diagnosis of A was confirmed previously only in post-hysterectomy cases and was thought to occur predominantly in patients over 40 years of age. Improved imaging techniques clearly show that younger patients also have A [22]. Laparoscopic identification of E with histopathological confirmation has long been accepted as the gold standard for E diagnosis, but recent guidelines support a non-surgical (clinical) diagnosis based on symptoms and imaging [23,24].

The differential diagnosis of E and A is guided by the postoperative histopathological examination of suspicious lesions. The diagnosis of malignant lesions is established following clinical suspicion and histopathological confirmation by identifying the primary structures affected by E or A, the malignant process and the benign–malignant tumor transition zone. The differential diagnosis is made in the absence of other neoplasms with secondary findings [14,25].

The treatment of E lesions can include medication with hormonal treatments: oral contraceptives (OCs), progestins (such as medroxyprogesterone acetate, GnRH analogues) and excisional surgery. The surgical approach is the most direct method of diagnosis. It requires advanced imaging assessment and examination of all potentially affected parts (abdominal wall, parietal peritoneum, visceral peritoneum, bowel, appendix, rectum, uterus, uterine torus, uterine ligaments, recto-vaginal septum, ovaries, fallopian tubes, bladder, ureters). Surgery has a curative potential, but the recurrence of lesions has been frequently reported [14].

### 1.4. Objectives

The potential for the preneoplastic or malignant transformation of E or A foci is a fiery topic in the field, with significant future applicability, especially in cases with significant clinical or histopathological characteristics. Thus, in this large study, our objective was to analyze the clinical and local factors (inflammatory, vascular, hormonal and the presence of oncogenic proteins) involved in the transformational process of E foci.

## 2. Results

All 243 cases analyzed in this study were divided according to location. Of the total number of cases, the control group (CG) included 24.70% who were victims of car accidents. These patients had no associated uterine pathologies. With family consent, uterine tissue was collected and processed for histopathological study. Half of the control individuals presented the endometrium in the proliferative phase (EPP) and half presented the endometrium in the secretory phase (ESP). Of the total cases, the pathological group (PG) of E/A represented 75.3%. These were cases diagnosed clinically with imaging or by undergoing a surgical and excisional treatment, depending on the localization. The excision of endometriomas was performed in cases with ovarian localization; the excision of E was performed in the cases with intraabdominal or parietal localization foci; the excision of lesions and colonic wall repair was performed in cases of colonic localization; and total hysterectomy was performed in selected cases with A.

The number and percentage distribution of cases studied from CG and PG and the incidence of pathology location in relation to the female population of reproductive age is detailed in Table 1. Referring to the overall incidence of E in the reproductive-age population, we have performed power analysis with type 2 estimation, significance level 0.05 and desired power 80% and achieved a sample size of 28,256.

In order to conduct the statistical analysis for this study, we first had to perform data screening to verify whether the prerequisites, data normality distribution and equality of variances were fulfilled. The sample size was above 30 cases; therefore, taking into account the Central Limit Theorem and the Large Enough Sample Condition, we can state that the data describes a “normal” behavior. In the t-table, the value of t becomes approximately equal to the z statistic around 30 degrees of freedom; hence, our assumption is correct.

Regarding the equality of variances, since we have almost equal number of observations in our sample, we can safely presume that the variances are equal and proceed with the statistical analysis.

### 2.1. Associated Symptoms Analysis: Pain, Infertility, Vaginal Bleeding and Bowel Disorders

The most common symptoms related to cases of E or A are the chronic pelvic or local pain (P), vaginal bleeding—metrorrhagia (Met) or menorrhagia (Men)— and bowel disorders (BD), and both conditions are frequently associated with primary (PI)/secondary infertility (SI). In this study, we observed that 25.10% of the patients had Met, 27.57% had Men, 18.52% had PI, 16.46% had SI, 18.52% of the total patients had BD and all patients had P with different intensity according to the visual analogue scale (VAS) of pain (Table 2). We assessed the severity of pelvic pain using the VAS (visual analogue scale), and the results were noted by category in Table 2.

We grouped the EPP and ESP cases and created a control group (CG) and a pathological group (PG) in order to perform statistical tests between symptoms. Analyzing each two symptoms grouped according to symptomatic control group (CG+), asymptomatic control group (CG−), symptomatic pathological group (PG+) and asymptomatic pathological group (PG−), we obtained the following data: we observed differences between Met and SI [X^2^(3, 486 = 57.29), *p* < 0.05], between Met and VAS 0->1 [X^2^(3, 486 = 106.991), *p* < 0.05], between Met and VAS 2->3 [X^2^(3, 486 = 19.048), *p* < 0.05], between Met and VAS 4->5 [X^2^(3, 486 = 8.479), *p* < 0.05], between Met and VAS 8->10 [X^2^(3, 486 = 9.291), *p* < 0.05], between Men and SI [X^2^(3, 486 = 9.627), *p* < 0.05], between Men and VAS 0->1 [X^2^(3, 486 = 92.600), *p* < 0.05], between Men and VAS 2->3 [X^2^(3, 486 = 60.605), *p* < 0.05], between Men and VAS 4->5 [X^2^(3, 486 = 9.946), *p* < 0.05], between Men and VAS 6->7 [X^2^(3, 486 = 487.020), *p* < 0.05], between Men and VAS 8->10 [X^2^(3, 486 = 12.240), *p* < 0.05], between PI and VAS 0->1 [X^2^(3, 486 = 297.748), *p* < 0.05], between PI and VAS 2->3 [X^2^(3, 486 = 60.559), *p* < 0.05], between SI and VAS 0->1 [X^2^(3, 486 = 55.980), *p* < 0.05], between SI and VAS 2->3 [X^2^(3, 486 = 57.703), *p* < 0.05], between SI and VAS 4->5 [X^2^(3, 486 = 150.641), *p* < 0.05], between SI and VAS 6->7 [X^2^(3, 486 = 11.306), *p* < 0.05], between BD and VAS 0->1 [X^2^(3, 486 = 62.279), *p* < 0.05], between BD and VAS 2->3 [X^2^(3, 486 = 60.559), *p* < 0.05] and between BD and VAS 8->10 [X^2^(3, 486 = 557.301), *p* < 0.05]. Examining the categories according to symptoms, we observed cases associated with Met ([Fig ijms-25-01789-ch001]A), cases associated with Men ([Fig ijms-25-01789-ch001]B), cases associated with PI ([Fig ijms-25-01789-ch001]C), cases associated with SI ([Fig ijms-25-01789-ch001]D) and cases associated with BD ([Fig ijms-25-01789-ch001]E).

### 2.2. Microscopic Characteristics Involved in the Development and Transformation of E and A Foci

We processed all 243 surgically excised tissue specimens by a paraffin-embedding technique using classical HE staining and special immunohistochemical staining techniques to highlight epithelial characteristics, the presence of perilesional inflammatory processes, perilesional neovascularization, hormonal receptors, the presence of perilesional stroma and the involvement of tumor proteins in cell proliferation.

#### 2.2.1. Microscopic Characteristics in Classical Staining

Ectopic foci of E have an epithelial, glandular structure, most often single layered in cases without hyperplasia, which preserves the functions of the normal endometrium. The peri-glandular stroma presents round or oval cells with reduced cytoplasm and central nuclei, as well as elements of the inflammatory system that maintain a proinflammatory microenvironment and neoplastic vessels. The endometrial mucosa used for comparison showed straight tubular glandular structures in EPP, lined by a simple columnar epithelium, and in ESP, the glands became sinuous, dilated. The normal endometrium laid on a basal membrane separating the two elements of the mucosa, the epithelium and chorion. The chorion showed numerous young cells—fibroblasts, lymphocytes, macrophages—blood vessels, collagen and reticulin fibers as well as endometrial glands in different stages of development.

In cases with A, the endometrial glands that extended beyond the uterine mucosa and implanted into the myometrial structure were surrounded by stromal cells (Figure 1A). In cases associated with hyperplasia with or without atypia, we found a layered glandular epithelium, marked local inflammation and changes in the nuclei/cytoplasm ratio. In cases of A associated with malignant transformation (A-EC-G1/G2/G3/CACC), we identified marked cellular atypia and an inflammatory process, the presence of necrosis and also newly formed vessels. In cases of OE, we identified endometrial glands with stroma in the ovarian cortex, often cystically dilated with marked perilesional inflammation; in cases of OE-A, we identified cellular atypia. In cases associated with DIE, we identified the ectopic presence of endometrial glands in the peritoneal structure, with stroma and inflammatory infiltrate in the mesothelial structure or in the sub-mesothelial connective tissue (Figure 1B). AE without atypia was identified in surgical excision specimens most commonly invading the rectus abdominis muscle, with endometrial glands and stroma visible in the entire muscular tissue. In cases that involved the intestinal wall, we observed the presence of endometrial glands and stroma all the way to the submucosa (Figure 1C).

#### 2.2.2. Microscopic Characteristics in Immunohistochemical Staining

The E or A foci, under the influence of the proinflammatory microenvironment and molecules secreted by inflammatory cells (cytokines, interleukins), transformed into precancerous lesions, with marked hyperplastic transformation observed in 4.12% of cases with A-H and in 3.29% of all cases represented by AE-H. In 2.06% of all cases, hyperplastic foci of A showed cellular atypia (A-AH); 1.23% of all cases showed endometriotic cyst atypia (OE-A). Malignant transformation was observed in 1.23% of cases with A-EC-G1, 2.06% with A-EC-G2, 2.06% with A-EC-G3 and 1.65% with A-ACC among all cases included in the study. The identified cellular types showed moderate to severe pleomorphism with epithelial layering, even forming microscopic papillae. These architectural changes also led to malignant transformation of the E and A foci. In our study, we observed that 12.76% of the total cases analyzed were associated with preneoplastic changes, and 7% of the total cases were associated with malignant lesions. The most common associated malignant subtype was endometrioid carcinoma.

##### Cytokeratin Reactivity Analysis

By using immunolabeling techniques with anti- CK7 and anti- CK20 antibodies, we were able to show that the profile of E/A foci matched the CK7+/CK20− profile and that these lesions represent metastases of tumors with a digestive origin. The exception to this pattern is the C-DIE cases, which show E foci corresponding to this CK7+/CK20− profile and presenting colonic glandular structures with a CK7−/CK20+ profile (Figure 2A). Depending on the intensity of the immunostaining, we classified the E/A lesions as strongly positive (+++), moderately positive (++), weakly positive (+) or negative (−−−), (Table 3). Thus, we observed that the intensity of the CK7 immunolabeling signal decreased proportionally with the lesion progression and tumor grading, resulting in intracellular molecular changes that disrupted the intermediate filaments of epithelial cells (Figure 2B).

##### Hormone Receptor Reactivity Analysis

The distribution of ER receptors and PR receptors varied according to the location of E/A pathology (Figure 3A,B, Table 4). We thus observed that the most variable lesions were A associated. The presence of these hormone receptors once again showed the endometrial origin of E foci.

##### Aspects of Endometrial Stroma at Ectopic Sites of E

We used anti-CD10 antibody to identify normal endometrial stroma in all types of A or E foci and support the diagnosis, as it is a very sensitive and useful immunohistochemical marker. Therefore, we observed that all cases in the study reacted positively for this immunostaining (Figure 4), and the peri-glandular stroma was detected.

##### Adjacent Stromal Vascularization Analysis

We used the anti-CD34 antibody to immunolabel the endothelium of the capillary vessels of neoformation (Figure 5A) and observed that the perilesional vascular density/mm^2^ increased in correlation with the incidence of hyperplastic foci (Figure 5B), the detection of cellular atypia and the presence of malignant transformation of E foci. Thus, we observed that the mean CD34 vascular density, standard deviation (SD) and post-hoc comparisons using the Tukey HSD varied according to the location of this pathology (Table 5, [Fig ijms-25-01789-ch002]). We observed that the highest vascular density was noted in the case of A-CACC, followed by A-EC-G3, A-EC-G2, AE-AH, AE-H and A-EC-G1, with the lowest vascular density in the case of OE. The number of vessels revealed the overall differences between the studied groups [F(15,242) = 450.601, *p* < 0.05].

##### Analysis of Adjacent Stromal Inflammatory Changes

We used the anti-CD3 antibody for T-lymphocyte immunolabeling (Figure 6A) and observed that peri-glandular cell density increased in proportion to the incidence of hyperplastic or even malignant foci (Figure 6B). Therefore, we observed that the mean CD3 cellular density, standard deviation (SD) and post-hoc comparisons using the Tukey HSD varied according to the localization of the lesions (Table 6, [Fig ijms-25-01789-ch002]). We noticed the highest cell density in AE-AH, followed by A-CCAC, and the lowest cellular density in OE. The number of cells revealed the overall differences between the studied groups [F (15,242) = 1970.346, *p* < 0.05] (Table 6, [Fig ijms-25-01789-ch002]).

We used the anti-CD20 antibody for immunostaining B lymphocytes (Figure 7A) and observed that the peri-glandular cell density increased in proportion to the occurrence of hyperplastic or even malignant foci (Figure 7B). Therefore, the mean CD20 cellular density, standard deviation (SD) and post-hoc comparisons using the Tukey HSD varied according to the location of this pathology (Table 7, [Fig ijms-25-01789-ch002]). We noted that the highest cell density was observed in AE-AH, followed by AE-H, and the lowest vascular density was observed in OE. The number of cells revealed the overall differences between the studied groups [F (15,242) = 459.200, *p* < 0.05].

We used the anti-CD68 antibody to observe macrophage immunolabeling (Figure 8) and observed that the peri-glandular cell density increased proportionally to the presence of hyperplastic and malignant foci. Thus, the mean CD68 cellular density, standard deviation (SD) and post-hoc comparisons using the Tukey HSD varied according to the localization of the lesions (Table 8, [Fig ijms-25-01789-ch002]). We noted that the highest CD68+ cell density was recorded in A-EC-G3, followed by A-EC-G1, and the lowest vascular density was observed in EPP. The number of cells revealed overall differences between the studied groups [F(15,242) = 417.152, *p* < 0.001].

We used the anti-tryptase antibody for mast cell immunolabeling (Figure 9A) and we observed that the peri-glandular cell density increased in relation to the presence of hyperplastic or even malignant foci (Figure 9B). Thus, we determined that the mean tryptase cellular density, standard deviation (SD) and post-hoc comparisons using the Tukey HSD varied according to the location of the lesions (Table 9, [Fig ijms-25-01789-ch002]). We noted that the highest tryptase+ cell density was recorded in the case of A-EC-G1, followed by A-EC-G2, and the lowest vascular density was seen in the case of EPP. The number of cells revealed the overall differences between the studied groups [F (15,242) = 457.346, *p* < 0.05].

##### Analysis of Cell Proliferation and Involvement of Tumor Proteins

We used the anti-Ki67 antibody to immunolabel dividing glandular cells and obtained the cell proliferation rate for every studied case. Thus, the percentage of Ki67-positive cells and the post-hoc comparisons using the Tukey HSD varied according to the location of this pathology (Table 10). We observed that the percentage of dividing cells increased with the occurrence of cellular atypia and malignant transformation (Figure 10A). The Ki-67 mitotic index showed global differences between the studied groups [F(5, 149) = 1232.019, *p* < 0.05].

To analyze the involvement of tumor suppressor proteins in the transformation process, the tissues were stained with antip53 antibody. We observed that the percentage of immunolabeled cells increased with the presence of cellular atypia and malignant transformation (Figure 10B). We observed the highest percentage of p53+ cells in the case of UE-EC-CACC, followed by UE-EC-G3, while in the cases of EPP, ESP, A, OE, AE, DIE and colorectal DIE, the reaction was negative (Table 11). The p53+ cell index showed global differences between the studied groups [F(8, 47) = 55.908, *p* < 0.05].

Moreover, with anti- BCL-2 antibody, we immunolabeled cells with potential to progress to malignancy by expressing this modified protein (BCL-2), which prevents programmed cell death (Figure 11A). Using immunolabeling techniques, we observed that all cases varied according to Table 12.

Similar to the anti-p53 antibody is the anti- PTEN antibody, which immunolabels cells with an activated tumor suppressor gene involved in cell-cycle regulation and the prevention of accelerated cell division. We observed that this gene was overexpressed in the case of atypia and especially in cases of malignant transformation of the endometrial glandular cells (Figure 11B). By immunolabeling, we observed that all cases varied according to Table 13.

Analyzing the percentage and numerical values of the inflammatory, vascular and proliferative elements in association with the symptoms, we obtained multiple correlations, as follows: EPP and ESP control cases, OE, OE-A, DIE and colorectal DIE cases were associated with lower mean values of vascularization and perilesional inflammation; however, the symptomatology was more accentuated in OE, OE-A or DIE, colorectal DIE cases compared to that in control cases. The most intense inflammatory process and vascularity were observed in cases associated with uterine lesions: A, A-H, A-AH, A-EC-G1/G2/G3/CACC, and in cases of AE, AE-H, AE-AH, but the symptoms are more marked in cases with a uterine site of the lesions.

Also, the percentage of dividing glandular cells was higher when the presence of atypia or tumor grading was higher, as well as the immunoreactivity towards tumor proteins p53, BCL-2 or PTEN.

##### Reported Trends between Symptomatology and Immunohistochemical Aspects

We observed that in A-CACC, where the highest density of CD34+ vessels was present (1513.48 ± 115.15 vessels/mm^2^), Met and Men were present in all cases with A-CACC ([Fig ijms-25-01789-ch003]A,B). We observed that the numbers of CD3+, CD20+, CD68+ and tryptase+ inflammatory cells tended to increase in line with the number of patients presenting with PI or SI, especially in cases associated with A-EC-G1 ([Fig ijms-25-01789-ch003]C,D). The number of patients presenting with BD was associated with an increasing trend in the number of neoformation vessels and the number of inflammatory cells in cases associated with A-EC-G1 and A-EC-G3 ([Fig ijms-25-01789-ch003]E). Mean CD68+ macrophages were highest in A-EC-G3 (2235.19 ± 98.88 cells/mm^2^) and coincided with the highest percentage values of Men and Met (100%, 100%). The cellular mean of tryptase+ mast cells was highest in A-EC-G1 (736.32 ± 40.45 cells/mm^2^) and coincided with the highest percentage values of PI (66.67%). We thus observe that CD34+ vessel counts and CD68+ macrophage counts were most frequently associated with Men and Met.

Also, in terms of pain, we observed that there was a directly proportional relationship between A-AH and VAS 0->1 ([Fig ijms-25-01789-ch004]A); VAS 2->3 tended to increase simultaneously with the increase in inflammatory cell density and neovascularization, especially in the case of AE ([Fig ijms-25-01789-ch004]B); VAS 4->5 tended to increase simultaneously with increased inflammatory cell density and neovascularization, especially in cases associated with AE-H ([Fig ijms-25-01789-ch004]C), but showed an inversely proportional relationship with OE and OE-A; VAS 6->7 had a tendency to increase simultaneously with increased inflammatory cell density and neovascularization, especially in cases associated with A-CACC ([Fig ijms-25-01789-ch004]D), and showed an inversely proportional relationship with OE, OE-A, DPE and DICE; VAS 8->10 had a tendency to increase simultaneously with increased inflammatory cell density and neovascularization, especially in cases associated with A-EC-G1 ([Fig ijms-25-01789-ch004]E), and showed an inverse proportional relationship with DPE and DICE. In the case of CD3+ lymphocyte presence, the highest cell average was present in the AE-AH cases (3762.14 ± 230.50 cells/mm^2^) and coincided with the highest percentage value of VAS 6->7 (40%) of patients. Mean CD20+ lymphocytes were highest in AE-AH (1100.34 ± 103.61 cells/mm^2^) and coincided with the highest percentage values of VAS 4->5 (40%) and VAS 6->7 (40%). CD3+ T lymphocytes were associated with pain at VAS 6->7; B 20+ lymphocytes were associated with pain at VAS 4->5; and mast cells were associated in our study with PI.

## 3. Discussion

### 3.1. Symptoms Involved in the Diagnosis of E and A

The most often associated symptom of E/A foci both in our study and in other research was P, regardless of location, being a symptom as complex as the disease itself, and it is well agreed that there is no correlation between P and the extent of E [26]. P can be nociceptive, hyperalgesic or a combination of these through central sensibilization, and E can create an environment suitable for the development of this symptom [27]. Also, hormonal status and coping mechanisms are known to influence pain perception [28]. Studies have shown that in DIE and C-DIE, the perception of P is more accentuated by higher nerve fiber density compared with other locations [29,30,31,32]. Our research found that in DIE and C-DIE, more than 40% of patients experienced severe P and very severe P. However, there is no well-established correlation between nerve fiber density and P severity according to the pain scale so far. Regarding the eutopic endometrium of E patients, a positive correlation between high nerve fiber count and P severity has been demonstrated [33,34]. However, CG was correlated at a low percentage with P in our study, most commonly mild or moderate.

Met and Men were also commonly associated with E/A, and studies have shown a correlation between these symptoms and an acquired inhibitor against factor VIII (co-factor in the blood coagulation pathway) in a case of severe pelvic E [35]. Therefore, when a patient with E has abnormal bleeding, we should look for indirect effects of E and several other possible causes. In OE, when chocolate cysts become large enough to apply strong pressure on the remaining ovary, it can no longer respond to stimulation or complete the ovulation process. Eventually, this leads to abnormal hormone production and possible abnormal ovarian bleeding [36]. The ovaries may be encapsulated by adhesions arising in E. DIE irritates adjacent tissues, and adhesions form through the initial inflammatory process of the organ to block the development of the E foci. When the ovary is stuck in adhesions, cysts may cause abnormal hormone production, which can lead to Met, Men [36].

P, Met and Men can cause extreme anxiety and stress to the patient [36], and they produce alterations in hypothalamic secretion, leading to a disruption of the function of many organs and the exacerbation of E/A-associated symptoms (P, Men, Met, BD, PI, SI).

DIE can present severe symptoms because the lesions penetrate deeper into the peritoneum and thus cause more pain than superficial lesions. DIE also tends to involve the uterine ligaments, uterine torus, rectum or vagina. The multifocal nature of such widely distributed lesions predisposes patients to a variable clinical presentation. The relationship between E and BD has not yet been fully elucidated, and several theories have been proposed. One of these theories is the immunological link through increased mast cell activation observed in both conditions, predisposing patients to the development of irritable bowel and BD. Patients with C-DIE show visceral hypersensitivity caused by the proinflammatory microenvironment generated, which contributes to the severity of gastrointestinal symptoms [37,38]. Current research has shown that four fifths of patients diagnosed with C-DIE have BD.

The relationship between E and infertility has been debated for many years. In the case of female fertility, it has been suggested that 30–50% of those diagnosed with E are infertile [39]. Despite extensive research, no agreement has been reached, and several mechanisms have been proposed to explain the association between E/A and infertility. These include distorted pelvic anatomy, endocrine and ovulatory disorders, altered peritoneal function, altered hormonal and cell-mediated functions in the endometrium, autocrine and paracrine factors, growth and transcription factors, cell adhesion molecules, immune and inflammatory mediators and other factors, including myometrial contractility and uterine peristalsis [40]. Abnormal uterine contractility may interfere with embryo adherence and penetration of the decidualized endometrium, representing a molecular trigger leading to infertility through these E-associated pathophysiological changes [41]. Based on the intraoperative aspects observed, distortion of pelvic anatomy, the so-called “pelvic factor”, may explain infertility in patients with severe forms of E more easily. In our study, we observed that IP and SP were associated with the presence of an accentuated inflammatory process and neovascularization process leading to lesion extension, especially in cases associated with A-EC-G1. These abnormalities may alter endometrial receptivity and embryo implantation, leading to PI or SI. Santulli et al. showed that the incidence of miscarriage in women with E is increased [42], also confirmed by recent systematic reviews and meta-analyses [43,44].

All these symptoms lead to the suspicion of E/A, and the physician should assess the pain (intensity, repercussion) and look for suggestive symptoms and the location of E. A patient’s report of multiple symptoms suggestive of E correlates with a greater likelihood of establishing a proper diagnosis [45,46].

### 3.2. Microscopic Characteristics of A and E, Local and Glandular Factors Involved in Their Transformation

Studies carried out in recent years have focused on the diagnosis and analysis of lesions of E and A. Thus, the identification of microscopic, pathophysiological and molecular factors involved in the process of development, evolution and transformation of E or A are under extensive research, and the molecular basis involved may be a therapeutic target in the future [7,47,48]. According to the World Health Organization (WHO), E is considered a potential precancerous lesion, and the genetic involvement is significant. Common genetic changes, such as those in PTEN and p53, are involved in the process of malignant transformation. The pathways that trigger the malignant process are still unknown, but a future topic of research would be to find the trigger factors and develop targeted gene therapies [49]. Further prospective studies with larger groups of patients are needed, as we cannot draw a conclusion from a relatively small sample size and still unclear pathogenesis [50].

Two hypotheses were described explaining the relationship between E/A and malignant transformation: the first described the presence of altered gene structures that allow endometrial cells to implant ectopically, undergoing malignant transformation, and the second is based on molecular changes in immune function that allow E/A to develop and increase the susceptibility of patients to malignant transformation [51].

Săndulescu, M.S et al. observed that the secretion of ectopic endometrial glands into the ovarian cortex led to compression and an increase in PI or SI [52]. We also noted similar findings in this study on EO. We also observed that some cases showed atypical cellularity in the ectopic glandular epithelium. The presence of these transformations may exist in isolation or may coexist with malignant ovarian tumor sites. The most common associated malignancies are clear cell carcinoma or endometrioid carcinomas [53].

The rate of cases diagnosed with AE has increased significantly over the last decade through the increase in the number of C-section deliveries performed. Intraoperatively, small epithelial structures can implant outside the pelvis, in the rectus abdominis, aponeurosis, umbilicus or intraperitoneal structures. Local inflammation arising from the ectopic presence of glandular structures, glandular secretion and compression exerted by cyclic changes has led to the development of a favorable microenvironment for the development of neovascularization of E foci, leading to their accelerated development with the impairment of patients both physically and psychologically by increasing P intensity [14]. The conditions created by the perilesional environment lead to the development of glandular hyperplasia and cellular atypia in the cases associated with the abdominal wall area.

Ismiil, N.D et al. and Motohara, K et al. described that outbreaks of A-H, A-AH or different types of E can be associated with EC or CACC [54,55].

This pathology is rare, with a 1% incidence of cases associated with malignant transformation reported in the literature [18,19,20]. The incidence of these transformations could be influenced by the missing histopathological examination of the whole tumor detected. A limitation of microscopic studies is that serial sections and serial analysis of stained tissues are not routinely performed to truly rule out the existence of an E/A site associated with malignant transformation. Also, a more extensive study would be needed in which several sections chosen randomly from the tumor would be analyzed. These are diagnostic blocks, and we could not exhaust the entire tissue. A future research direction would be to make the process of tissue sectioning and staining automated, along with a serial histopathological analysis and a 3D reconstruction of E/A foci. The 2D analysis of the sections offers certain limitations regarding the evaluation of the entire lesioned area, both the central area and the “invasive front”. Serial sectioning and 3D analysis of endometriosis foci would certainly provide much better data. In this way, both entire tumor formation and the invasion front can be accurately analyzed. Full scanning of sectioned tissues can allow for digitization of the histopathological process by generating whole-slide images (WSI) and facilitating histopathological analysis. Recent data have demonstrated a strong correlation between the underlying tumor molecular profile and morphological patterns obtained from WSI images of tumor tissues [56]. Cifci et al. provided analytical details with a comprehensive review of recent work (e.g., papers published between 2017 and 2021) on deep learning to demonstrate that genetic alterations in tumor tissues are predictable using histopathological imaging with artificial intelligence (AI)-based methods. Also, Rakha et al. supported that automation and the use of AI would emphasize a direct correlation between morphological features, such as tumor grade and types, as well as clinical behavior and response to therapy [57]. A is a benign pathology, but it has some characteristics similar to malignancy: it proliferates, invades and creates a molecular background that determines the presence of neovascularization with accelerated growth of foci and increased invasive potential. The literature has described the transformation of A, the presence of hyperplasia, cellular atypia and subsequent malignancy to varying degrees. These transformations were conditioned by the presence of molecular substrate and the genetic activation of oncoproteins or deactivation of proteins that regulate programmed cell death [58]. The malignant transformation of E/A has been claimed to be associated with metaplastic transformation, with a proinflammatory tissue reaction created by the distortion of tissue architecture and the role of mechanical and hormonal factors. Myocytes adjacent to A foci respond to mechanical and hormonal changes caused by local factors by remodeling the actin cytoskeleton, leading to hypertrophy and hyperplasia of A foci [58]. Studies in this area showed that A development and transformation are conditioned by the ability of proliferation, differentiation, cell apoptosis, angiogenesis and protease production, leading to changes in the expression of genes encoding stromal cell transformation. However, so far, no clear evidence supports genetic or epigenetic involvement in E/A malignant transformation [58].

#### 3.2.1. The Role of Cytokeratin

Classical HE staining identifies the presence of ectopic glandular structures but cannot differentiate them from possible metastases with a different point of origin. Thus, by using immunohistochemical studies using anti-CK7 and anti-CK20 antibodies, we could make a clear diagnosis of E/A. CK7 is found in the structure of the glandular endometrial epithelium [59,60], and CK20 is found in the structure of the intestinal epithelium [48]. CK7 is also used for differential diagnosis with a digestive site tumor, where immunostaining shows CK7−/CK20+ [48]. CK7 is also positive in the hyperplastic or malignant transformed epithelium, but studies have shown that the intensity of immunoreactivity is in an inverse relationship to tumor grading [14,48]. CK20 is a type I cytokeratin and is positive in adenocarcinomas of the gastrointestinal tract and non-mucinous ovarian adenocarcinoma [61].

Double immunostaining with the two types of cytokeratin completes the differential diagnosis of tumors with endometrial or digestive origin. In our study, CK7 was positive in all A/E cases and negative in the intestinal epithelium, and CK20 was positive only in the intestinal epithelium of colorectal DIE cases.

#### 3.2.2. The Role of Hormone Receptors

We observed that immunoreactions for anti-ER and anti-PR antibodies varied according to the presence of E/A. Thus, ER was present in all E lesions and absent from some cases of A, A-H and A-CACC. PR was present in all E lesions except for 10% of A cases.

Our study showed that normal endometrium, EPP and ESP, showed ER in all cases, similar to the studies performed by Apostolou, G. et al., which claimed that ER is present in the eutopic endometrium, and in the ectopic endometrium, there may be changes in the ability to recognize sexual hormones and escape their control [62].

Similarly, we observed that PR was present in all cases of E and absent in 10% of cases of A. The presence of ER and PR in E/A shows that these structures are hormonally dependent to varying degrees. This is demonstrated by the cyclicity of E-associated signs and symptoms and by the favorable response of this pathology to hormonal therapy [63].

#### 3.2.3. The Role of Perilesional Vascularization

The microenvironment created by E/A foci, rich in inflammatory cells secreting cytokines, interleukins and proangiogenic factors, leads to the activation of the neovascularization process that contributes to tissue growth through cell proliferation and the development of A/E foci and supports the possibility of the vascular dissemination of endometrial cells [48]. CD34 may act as an adhesion molecule to T lymphocytes and contribute to their mobilization to lymph node stations [64]. It can also adhere to eosinophils or mast cells, blocking their action [64,65]. We observed that the highest perilesional vascular density was preexisting in A-EC-G1/2/3, followed by A-CACC, A and AE, similar to previous studies [14,66], and contributes to the accelerated growth of E/A foci and the development of severe symptoms.

#### 3.2.4. Expression of Endometrial Stromal Cells

CD10 expression has been demonstrated in a wide range of normal or transformed malignant cells. The literature data demonstrated that CD10 expression can be lost in a subset of tumors associated with malignant transformation, and this loss is correlated with the acquisition of tumor-promoting properties. It has also been shown that the loss of CD10 expression supports the development of CACC [67]. In this study, CD10 immunolocalization was cytoplasmic in endometrial stromal cells in all cases included in the study, except for 12.35% of cases that represented A, which lacked the presence of peri-glandular stroma and thus immunoreactivity for CD10.

#### 3.2.5. The Role of Tumor Proteins in E/A Transformations and Their Contribution to Cell Proliferation

Continuing the previous study of Istrate-Ofiţeru, AM et al., in detecting the involvement of oncoproteins in E/A transformation, we used anti-p53, anti-BCl2 and anti-PTEN antibodies. For the detection of the proliferation extent of endometriotic fungal cells, we used anti-Ki67 antibody, which immunolabeled the dividing nuclei [14]. We analyzed the immunoreactivity of these antibodies in endometrial glands and we observed that with the appearance of transformation, the reactions were positive and the degree of cell proliferation increased.

Alterations in the tumor suppressor protein p53, cited in the literature as a “genome guardian”, are involved in inhibiting cell apoptosis, inducing angiogenesis and disrupting genome stability. p53 contributes to repairing damage to the deoxyribonucleic acid (DNA) structure by blocking the cell cycle at certain phases, so that there is enough time for the protein to repair the damage [68]. In our study, we observed that an alteration in the tumor protein was present in cases associated with the presence of cellular atypia or in cases associated with malignant transformation, the highest percentage being recorded in the case of A-CACC, followed by A-EC-G3, A-EC-G2 and the lowest percentage being recorded in the case of A-AH. We observed that the most susceptible cases for the alteration of p53 function were those with uterine localization of the initial A lesions.

Regarding BCL-2, it is well known that this oncoprotein is involved in the regulation of cell apoptosis, being able to induce or inhibit it [69]. In the E/A foci analyzed, this protein was present, especially in cases with malignant transformation—A-EC-G3, A-CACC, A-EC-G2, A-EC-G1—with an intense positive reaction. Also, A-H, A-AH, AE-H, AE-AH and C-DIE had a moderate positive reaction for anti-BCL-2 antibody, and A, OE, OE-A, AE and DIE lesions had a weakly positive reaction to immunolabeling. Thus, we deduced that both uterine and deep, ovarian, intestinal or abdominal wall lesions can cause pre-stage BCL-2 oncoprotein alterations and can transform into premalignant or malignant lesions, and previous studies have shown that intraepithelial oncoprotein alterations cause a rapid and chaotic progression with potential for malignant transformation [70].

PTEN is a tumor suppressor protein that has been shown to be involved in the development of several types of cancer. It is involved in cell-cycle regulation via a protein phosphatase that prevents accelerated cell multiplication [71]. Mutations in it block the suppressor enzyme function, inhibit programmed cell death and lead to increased rates of cell proliferation. In our study, we observed that immunoreactivity was intensely positive in A-EC-G1, A-EC-G2/G3, A-CACC, AE-H and AE-AH. A moderately positive immunoreaction was also present in A-H, AE-H, AE-AH and DIE cases. A weakly positive immunoreaction was present in A, OE, OE-A and AE cases. The immunoreactivity was completely negative in the control groups and varied in intensity in the other cases, which supports the hypothesis that this gene may be involved in A or E transformation in any site.

Overlaying these data, we can argue that the most susceptible A/E lesions for transformation are those with uterine localization, followed by those at the abdominal and ovarian levels, showing immunopositivity for the three proteins in certain percentages. In contrast, DIE or C-DIE lesions showed immunoreactivity only for PTEN or BCL-2.

In order to investigate the involvement of these tumor proteins or oncoproteins in the evolution of E/A foci, we analyzed and calculated the cell proliferation rate of ectopic glandular structures. Therefore, we used the anti-Ki67 antibody, which is a marker of cell proliferation [71,72], to calculate the rate of cell division, which was increased when malignant transformations were present and when cell division became uncontrollable, similar to previously performed studies [73,74]. The highest percentage of dividing glandular cells was recorded in A-CACC, followed by A-EC-G3, A-EC-G2 and AE-AH, and the lowest percentage was recorded in DIE. We thus reinforce the idea that the most prone E/A lesions for malignant transformation are those at the uterine level, and although DIE lesions may mimic a malignant process by their polypoid character or by their ability to infiltrate adjacent tissues, they were seldom associated with the presence of mutations at the level of tumor suppressor proteins or with an increased cell division rate [14].

#### 3.2.6. Involvement of the Perilesional Inflammatory Process

It is well known that inflammation is a predisposing factor for cancer development and promotes all stages of tumorigenesis. Cancer cells, as well as surrounding stromal and inflammatory cells, engage in well-orchestrated reciprocal interactions to form an inflammatory tumor microenvironment (TME). Cells in the TME are highly plastic, continuously changing their phenotypic and functional characteristics. Thus, inflammation leads to tumor induction, growth, progression and metastasis. Also, local, perilesional inflammation maintains tissue homeostasis or contributes to tissue repair. Defining the involvement of the inflammatory process in the local molecular and cellular mechanisms that promote tumors is essential for the further development of anti-cancer therapies. Thus, inflammation can transform cells in E/A foci, triggering the appearance of cellular atypia with an altered nucleocytoplasmic ratio; nuclei can become hyper- or hypochromic, and cell layers can multiply chaotically. This inflammatory microenvironment associated with endothelial dysfunction participates in carcinogenesis [7].

Leukocytes, lymphocytes, macrophages and mast cells act by synthesized and secreted products on the normal, eutopic endometrium but mostly on ectopic foci of E/A, as observed in the current study.

CD3 was expressed in mature T cells identified at the peri-glandular and intraepithelial level by immunolabeling with anti-CD3 antibody. Excessive T-lymphocyte overgrowth supports a significant inflammatory process or pathology with an unfavorable, hyperplastic or even malignant course [75]. The highest CD3+ T-lymphocyte density was found in AE-AH, A-CACC and AE-H being followed by lesions such as A-EC-G3/G2/G1, and the lowest T-cell density was obtained around C-DIE foci. Thus, altered gene expression and CD3+ inflammatory infiltration, were defined as informative for the prognosis of malignant transformation and cell proliferation at E foci [7,76].

CD20 was expressed on the surface of B lymphocytes, with a role in cell interaction and supporting a proinflammatory and pro-tumorigenic microenvironment [77]. The highest B lymphocyte cell density was identified in perilesional AE-AH foci, followed by A-CACC and AE-H lesions, and the lowest CD20+ lymphocyte density was found in OE lesions. Analyzing these aspects, we observe that AE-AH-type lesions were associated in all cases with the presence of altered PTEN/BCL-2 and the highest CD3+/CD20+ cell densities, which supports the hypothesis that the proinflammatory and tumorigenic microenvironment supports the alteration of oncoproteins and contributes to the appearance of cellular atypia [56,77].

Monocytes and macrophages both from the blood flow and from tissue (microglia, Kupffer cells) express CD68. This molecule is a transmembrane glycoprotein that has been identified and studied over time in foci of DIE [78,79]. It has a role in mediating and exacerbating inflammation and supporting the microenvironment adjacent to E/A foci [80,81]. Research has shown that macrophages are able to synthesize and secrete proinflammatory mediators; cytokines; interleukins 1, 2, 6, 8, 10 and 22 (IL-1, IL-2, IL-6, IL-8, IL-10, IL-22); tumor necrosis factor-alpha (TNFα) [79,80,81]; macrophage-derived growth factors; T lymphocytes [81]; fibroblast growth factors and fibronectins; angiogenesis factors; endothelial vascular growth factor; platelet activation factors; and hepatocyte growth factor (HGF) [70,80,81], and all these factors provide the necessary environment for the development of E/A foci and their potential for growth and premalignant or malignant transformation [78,79,80]. Macrophages and neutrophils are potent producers of reactive oxygen species (ROS) and nitrogen (RNI) that induce mutations in the ectopic foci of E. The induction of inflammation can lead to mutagenesis, predisposing patients to epithelial mutations in p53, PTEN or BCL-2 [82,83,84,85] without the presence of other exogenous mutagens [86]. Macrophages play an essential role in the destruction, repair and regeneration of endometrial tissue during the endometrial cycle, especially in the menstrual and proliferative phases of the cycle [87], but in E/A foci, they can escape macrophage control [87,88].

Using anti-CD68 antibody, we demonstrated that the highest macrophage cell density was in cases associated with A-EC-G3/G1, A-AH and AE-AH, and the lowest macrophage density was in EPP, control ESP and DIE, contrary to the studies of Khan et al., who claimed that there was an increase in macrophage populations in both the EPP and ESP of the endometrial cycle in women with E [88]. A or abdominal wall locations were again most affected. Two categories of macrophages have been described over time in similar studies; the first type induces inflammation (M1), and the second type decreases inflammation (M2) and helps tissue repair. Both types of macrophages may be involved in E/A evolution, but the fact that atypical or malignant foci are associated with the highest number of macrophages suggests that the proinflammatory ones are more abundant [88,89]. Microscopic studies have shown that E/A may have increased numbers of perilesional macrophages [88] and increased amounts of monocyte chemotactic protein 1 (MCP-1), which is a β-chemokine that produces chemotaxis and macrophage activation [71,81,87,90]. Under the influence of all local factors, the ability of macrophages to phagocytose cellular debris decreased and the rate of endometrial proliferation increased, therefore increasing the rate of cell proliferation and the transformation of foci of E/A. M2 macrophages are involved in the progression of malignant transformation of E/A by increasing the invasive potential and inhibiting cell apoptosis [91].

In the control groups with EPP or ESP, and also in the DIE lesions, we found a decreased cell density of mast cells immunolabeled with anti-tryptase antibody, and their density was increased in cases associated with A-EC-G3/G2/G1, indicating that the local functional balance was disturbed. Mast cells produce a wide range of proteases and enzymes, the most frequently mentioned being tryptase, used as an immunohistochemical marker in cell identification [92]. It has been shown that mast cells are involved in the fibrogenesis process and can initiate a local inflammatory response by releasing several mediators such as proteases, histamines and cytokines: IL-1, IL-6, IL-8, TNFα, TGFβ, macrophage granulocyte colony-stimulating factor (GM-CSF) [92]. All these molecular compounds such as cytokines IL-1α, IL-6, IL-8, IL-18, and TNFα were increased in the presence of E/A and maintain a potent proinflammatory environment that can activate tumorigenesis [81,93,94]. We also found that mast cell density is higher in A-EC than in the other categories and that, according to other studies, there is a minimum number of mast cells in the normal, eutopic endometrium [95].

After an integrated and comprehensive analysis, this study shows that the panel of immuno-markers used to immunolabel inflammatory cells, oncoproteins, neovascularization, hormone receptors and dividing cells show a unique pattern of expression depending on the E location or depending on the presence of cellular atypia or malignant transformation.

## 4. Materials and Methods

This is a retrospective study, including 243 admitted patients (woman of reproductive age), investigated and surgically treated within the II Clinic of Obstetrics and Gynecology and the III Clinic of General Surgery of the Emergency County Clinical Hospital of Craiova, Romania; in the Obstetrics and Gynecology Clinic of the Clinical Hospital of Obstetrics and Gynecology “Prof. Dr. Panait Sarbu”, Bucharest, Romania; as well as in the Policlinico di Monza Hospitals, Bucharest, Romania. Also, control cases were obtained from patients who died in car accidents and whose relatives gave their consent for study participation. The victims were included in the database of the Forensic Medicine Institute of Craiova, Romania. Informed consent was obtained from all subjects or their legal representants. The patients or their legal representants signed a consent form for personal data use. The study was conducted according to the guidelines of the Declaration of Helsinki and approved by the Committee of Ethics of the University of Medicine and Pharmacy of Craiova, Romania (88/13 September 2018) and through project no. 26/531/14 on 31 May 2022.

Postoperatively, depending on the type of surgery, excision in cases of OE, OE-A, AE, AE-H, AE-AH, C-DIE or DIE; hysterectomy in the cases of A, A-H or A-AH; or radical in cases associated with malignant transformation: A-EC-G1/G2/G3/CACC, the tissue was harvested and placed in 10% formalin for fixation. Postoperative pieces were photographed and then sectioned for inclusion in formalin. Tissue fragments were sent to the Center of Microscopic Morphology and Immunology of the University of Medicine and Pharmacy of Craiova, Romania, to be processed and analyzed microscopically (director: Professor MD PhD Mogoantă Laurențiu, https://eeris.eu/ERIF-2000-000G-1574, accessed on 1 February 2019).

We analyzed 243 cases: 60 of cases in the CG (30 control cases with EPP, 30 control cases with ESP) and 183 cases in the PG distributed as follows: 62 cases with uterine sites (30 cases with A, 10 cases with A-H, 5 cases with A-AH, 3 cases with A-EC-G1, 5 cases with A-EC-G2, 5 cases with A-EC-G3, 4 cases with A-CACC), 33 cases with ovarian sites (30 cases with OE, 3 cases with OE-A), 43 cases with parietal sites (30 cases with AE, 8 cases with AE-H, 5 cases with AE-AH), 30 cases with DIE and 15 cases with C-DIE.

### 4.1. Analysis of Patient Symptoms and Infertility

All patients or their legal representants in cases of road traffic accident victims completed a form stating whether their pathology symptomatology was associated with included pelvic or abdominal wall pain, (pain scale: 0->1 no pain; 2->3 mild pain, 4->5 moderate pain, 6->7 severe pain, 8->10 very severe pain/excruciating), metrorrhagia, menorrhagia, primary/secondary infertility or bowel disorders.

### 4.2. Histopathological Analysis

The histology research facilities were provided by the Research Center for Microscopic Morphology and Immunology, University of Medicine and Pharmacy of Craiova (UMFCV). The tissue samples were acquired and diagnosed between 2018 and 2023. We performed the histological preparation and examination within the Histology Department of the UMFCV. After fixation of the biological material in 4% neutral buffered formalin, the tissue fragments were routinely processed for paraffin embedding and sectioning as 4 µm-thick sections [96].

Classical hematoxylin–eosin (HE) (Hämatoxylin-Lösung nach Mayer, Sigma-Aldrich-MHS16 MSDS; Eosin Y-Lösung, alkoholisch, Sigma-Aldrich-HT1101128 MSDS) staining revealed the presence of ectopic foci of E or A, but the final diagnosis was determined by immunostaining techniques, being able to make a differential diagnosis from other metastatic lesions.

The immunohistochemical protocol was based on a standardized algorithm, with variations depending on the specific antibody used.

First, for both classical HE staining and immunolabeling, plain or poly-L-lysine-treated slides were obtained on which fine tissue sections, 5 μm thick, were applied, obtained using the HMB350 water-based transfer micrometer. They were placed in xylene baths (3 baths × 15 min) for deparaffinization, followed by alcohol baths of decreasing concentration (100%, 90%, 70%, 5 min each) for dehydration, and then the sections were hydrated with distilled water (dH_2_O) (3 baths × 5 min). For classical staining, the cell nuclei were stained with hematoxylin and the cytoplasm with eosin, and for special staining, the cells were first submitted for antigen retrieval by microwave in a specific buffer solution of citrate pH 6 0.1 M or ethylenediaminetetraacetic acid (EDTA) pH 9, according to the manufacturers’ specifications (Table 14). Tissue structures were incubated in 3% hydrogen peroxide (H_2_O_2_) solution for 30 min to block endogenous peroxidase that might interfere with signal detection, and then in 3% skimmed milk saline solution (blotting grade blocker no fat dry milk by Bio-Rad, inventory number RBD1706404XTUEA) to block non-specific antibody binding sites. Next, after obtaining the appropriate dilution in bovine serum albumin (Table 1, Dako, Glostrup Denmark and Abcam, Cambridge, UK), primary antibodies were applied to slides and incubated for 18 h in the refrigerator at 4 °C. After this step, slides with tissue sections were left at room temperature and then washed with phosphate-buffered saline (PBS) and incubated with HRP-labeled secondary antibodies specific to the species of the primary antibodies (Nichirei-Bioscience, Tokyo, Japan). The last step of the reaction was the actual detection of the signal utilizing 3,3 ‘diaminobenzidine (DAB, Nikirei-Bioscience), after which cell nuclei were counterstained with H, dehydrated, clarified in xylene and covered with special slide-mounting medium—Canada Balm—for imaging and analysis.

### 4.3. Method of Analysis

Patient forms were filled in and categorized in Microsoft Excel 2010 according to the type of E/A associated with simple hyperplastic changes, atypical changes or even malignant transformation. Regarding the associated symptomatology, according to location, the chi-squared test was applied using the same program, and the values obtained were explained in the text.

In order to quantify vascular and inflammatory levels or the percentage of Ki67+, p53+ or immunolabeling presence for ER/PR/PTEN/BCL-2/CD10, we took four images with a 400× objective lens of each specimen using constant exposure and manual exposure and illumination configurations. We scored the vessels, perilesional cells or nuclei of the immunolabeled glandular epithelium, averaged the 4 values and plotted them. Two images captured the nucleus (“center”) of the lesion and 2 images captured the pro-invasive glandular lesion (“invasive front”) in E. The arithmetic mean was performed between the values obtained, regardless of the plotted area, in order to minimize the molecular discrepancies in the analyzed fields depending on the group.

Images were acquired using a Nikon Eclipse 55i microscope equipped with a 5-megapixel cooled CCD (Nikon DS-5Mc (Nikon Europe BV, Amsterdam, The Netherlands) and supported by the Image ProPlus AMS 7 image analysis package (Media Cybernetics, Rockville, MD, USA). We examined the images obtained and observed involvement of the inflammatory process in the unfavorable evolution of the transformed E/A foci. All elements (cells, blood vessels, dividing cells) were manually scored using the “manual labeling” function in Image ProPlus, averaged on each slide and then for each pathological panel group. To convert the numerical cellular value/400× field of view (FOV) area into cellular densities /μm^2^, we normalized the values obtained for 400× FOV to 1 mm^2^ considering the total FOV 40× area of 456 µm^2^.

Data were stored in Microsoft Excel sheets, and statistical analysis was performed in SPSS using univariate analysis of variance (ANOVA) with Tukey HSD (honest significant difference) post-hoc analysis with multiple comparisons (significance set at *p* < 0.05) for continuous data. The obtained values were organized in tables, including *p*-values. The chi-squared test was used for categorical data. In terms of the overall incidence of E, we calculated the power of analysis (sample size, type 2 error estimate). Charts present mean values and standard deviations (SD) for continuous values. For Ki67+ dividing nuclei and p53+ nuclei, we counted the total number of nuclei in the glandular epithelium and the number of positive nuclei and expressed them in percentages (%) using Microsoft Excel. Final percentages were determined. For PTEN and BCL-2 analysis, the cases were scored as (+/+/+++) and (−−−) and were represented as a percentage of the total number of cases with similar location.

## 5. Conclusions

This comprehensive study of the changes in the ectopic endometrial glandular epithelium, as well as the local factors, is important in order to establish the possibility of progression and transformation of E/A, as well as in setting standards for the prognosis and management of the disease.

The appearance of cellular atypia or malignant transformation associated with E/A structure can occur under the influence of the proinflammatory microenvironment, particularly in inflammatory cells, which provides a favorable environment for neovascularization and the presence of mutations in tumor suppressor proteins or oncoproteins, with an associated increase in cell proliferation and tumor growth. Also, the presence of hormonal stimulation promotes tumor development and establishes the correct treatment. All these local factors contribute to the enhanced invasiveness of E/A, with an exacerbation of the patient’s symptoms and the presence of imagistic markers, leading to the diagnosis of E/A and the likelihood for future therapeutic progress.

Increased perilesional vascularization maintains tumor proliferation and leads to compression with enhanced proinflammatory changes and alterations in local histoarchitecture. Proper investigation and detection of E/A in early stages offers suitable therapeutic options with a marked decrease in symptoms.

The detection of atypical or malignant areas of E/A by immunostaining techniques can reveal the extent of tumor proliferation, inactivation of tumor suppressor genes, blockage of programmed cell death, presence of proinflammatory microenvironment or enhanced neovascularization and can dictate a gene- or molecule-targeted treatment to block tumoral growth or transformation.

## Data Availability

All data presented here are available from the authors upon reasonable request.

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
