# Peer review of "Clinical Characteristics and Local Histopathological Modulators of Endometriosis and Its Progression"

_ijms, 2024, doi:10.3390/ijms25031789_

Round 1

Reviewer 1 Report (New Reviewer)

Comments and Suggestions for Authors

Brief comment

The present manuscript ijms-2784753 is an extension of 10.3390/ijms23105614 already published in IJMS. However, the previous report was somehow superior in presentation, precision and scientific soundness. 

Major criticisms

  1. Statistics have not been performed except at l190. The authors should at least perform statistics as in their previous reports: statements like « There was more A than B » should be demonstrated with valid statistical tests. Additionally, power of analysis should be calculated (sample size, type 2 error estimation) at l1253. Statistical tests and threshold for considering a p-val significant must be described with more details in the Method section. Consider the help of a trained statistician.
  2. Table2: patients from groups EPP and ESP show VAS 2->3 of 66.67 and 56.67, respectively. However, patients of these control groups were victims of car accidents (l146). Therefore, how the authors could collect VAS data from deceased patients? Adapt also l1255.
  3. Recent papers have demonstrated significant differences in topology and molecular expression between the core ("centre") and the progressive gland lesion ("invasive front") in endometriosis. Therefore: /3.1 Please justify how the 4 images have been selected for analysis. The authors must specify whether the analysed samples represented the core or the invasive front. /3.2 Please address the limitation that 2D sections only partially capture the 3D conformation and environment of the lesions.
  4. The use of references is the object of important criticism. For example, (1) reference 8 is incorrectly cited at l84, l92 and l103. Considering ref8, the authors must refer to the book or (preferred) to the chapter. (2) Two references dated 1985 and 1986 are given to introduce ESR1, a concept that is not discussed further. (3) The sentence l902 is copied directly from ref33. However, adenomyosis was not studied in the cited study, and the enzyme is not specified in the present manuscript. (4) Reference to Khan is 94, not 93 (l1185). Therefore, the referee strongly recommends using recent literature (typically 10 years), making judicious use of these references, and performing a complete check of the citations. Comment to the authors: while all claims must be supported by references, only relevant claims should be made. In other words, the referee strongly recommend simplifying the text, as the current manuscript is exceptionally long for an original research.
  5. In particular, the discussion is far too long (almost 8 pages !) and needs to be simplified: authors should not repeat their results, and only the most significant results should be discussed in concise language and compared with the current knowledge. The discussion should never contain new results or repetition. Every element required for the discussion should at least be introduced in the Introduction section.
  6. How the authors integrate their findings and discussion in section 4.3.4 with the report from Khan et al. on CD10 expression in adenomyosis 10.1016/j.rbmo.2019.03.210 ?
  7. Importantly, limitations and bias are not taken into account.
  8. Similarly, the manuscript misses perspectives and future direction. On a technological point of view, it could cover the use of AI and digital pathology on WSI.

Minor criticims

  1. Much of the introduction is written as specific to endometriosis. Adenomyosis is thus poorly presented. The differences between the two diseases are not indicated.
  2. The frequency of malignant transformation or endometriosis-associated malignant tumours is not indicated. Please refer to 10.1093/humupd/dmaa045 and  10.3390/diagnostics13111883
  3. The correct magnification factor for the objective lens must be 40x (and 10x for the ocular/camera mounting adapter, hence resulting in calculated 400x power magnification); please amend for precision. The reviewer discourage the use of the term "power magnification" as it does not reflect the analysed area (wide range of values for HPF given the microscope, see 10.4103/jpi.jpi_48_20). The reviewer recommends the use of a scale bar when possible, and a clear definition of the analysed FOVs. For density, please refer in cells per square mm.
  4. The sentences « Informed consent was obtained from all subjects. The patients signed a consent form for personal data use.» must be rephrased for deceased patients (l1236). See statement l1234 and l1254-1258 to adapt. Idem l1391.  
  5. Paragraph l1219-1226 can be deleted. The advantages of multiplexing are already well established in the scientific community and discussed in the previous publication of the authors (10.3390/ijms23105614). Conversely, only the advantages are discussed, not the limitations of multiplexing, so there is limited technological novelty here. Furthermore, considering multiplex IHC with brown-pink-violet chromogens as a new methodology is questionable. In fact, one may consider TSA technology superior, because it allows a greater number of combinations, is compatible with colocalization analysis and allows spectral unmixing. As for novelty, the authors should consider commercially available technologies that mitigate the technical limitations (labor-intensive, need of optimisation, epitope denaturation,…).
  6. Section l1373-1375 must be deleted.
  7. Provide reference of all the reactive and chemical compounds (e.g. Mayer’s Hematoxylin). Correct the spelling l1290.
  8. Legend to Figure1: « Morphological and immunohistochemical aspects of ^adenomyosis and^ endometriosis lesions ». The authors might consider dividing this figure, as done in their previous publication. Another option is to indicate the type of lesion on the top and the markers on the left, per quadrant. The use of a scale bar would be beneficial in any case. For reasons of unity, a standard arrowhead should be used.
  9. Charts showing percentage should stop at 100%. Do not use decimals here. Indicate also the number of cases in the plots (e.g. 0/4, 5/5 or 23/30) above the bars.
  10. Adapt Table1 as follows for simplifying reading the table and the legend: left column, use the full spelling of conditions (with abbreviation in parentheses); only one right column with the number of cases (and percentage in parentheses). 
  11. Adapt Table 2 as follows: indicate numbers (and percentages in parentheses).
Comments on the Quality of English Language

The English language must be rephrased for concision. 

Specific errors:

  1. Correct L763: « Met and Me^N were present in all cases »?
  2. Use of convention L932: Use the 'name of first author et al.' for starting a sentence, not reference in brackets. As done l938.
  3. Precise L1283: ^Bovine (?) skim milk.
  4. Use of abbreviations: CD, SD, ... must be defined once and used thoroughly, except for titles, figures, tables and charts.
  5. Inconsistent use of abbreviation (e.g. l400, endometriosis must be abbreviated). Check the manuscript thoroughly. 
  6. Use abbreviations in author contribution section, as given l7-27.

Author Response

The replies can be found in the attached document. Thank you!

Reviewer 2 Report (New Reviewer)

Comments and Suggestions for Authors

This is a large-scale study based on histochemical and immunohistochemical analysis of 243 endometriosis or adenomyosis tissue samples. This reviewer did not understand why the manuscript was presented with an activated track change, which penalized reading. This version of the manuscript was not presented as a previously revised version.

The main general comment is that there is excessive redundancy between the description of the results in the text and the presentation in the tables. This could be improved by deleting most of the detailed information on lines 157-172, 195-222, 320-329, 378-379.

The conclusion of the summary is not a direct conclusion of the study results.

It is unclear why the term "endometriotic" has been replaced by "endometrial" in lines 106, 119 and 129. The term "endometriosis" seems more appropriate than "endometrial".

In Tables 5-11, please explain the selection of the p-value p>0.005 in the last column on the right.

Line 447: Reference to Figure 4K should be replaced by Figure 1K

Line 492: Reference to Figure 1N should be replaced by Figure 1O

Line 630 : Reference to Figure 4U should be replaced by Figure 1U

Line 671: Reference to Figure 4V should be replaced by Figure 1V

Figure 1 should be presented earlier in the article

Line 749: "were" should be deleted

Comments on the Quality of English Language

Quality of English is correct

Author Response

The replies can be found in the attached document. Thank you!

Round 2

Reviewer 1 Report (New Reviewer)

Comments and Suggestions for Authors

The authors have satisfactorily addressed the comments of the referees: the revised manuscript has improved in precision and readability. The overall presentation has notably improved in terms of method description and figure quality. Importantly, the manuscript has gained in scientific relevance by implementing the statistical analysis of this large-scale study. However, it is unclear if the data distribution & homoscedasticity were respected for conducting parametric tests. 

Comments on the Quality of English Language

Some minor typos

Author Response

The response can be found in the attached Word document 

This manuscript is a resubmission of an earlier submission. The following is a list of the peer review reports and author responses from that submission.

Round 1

Reviewer 1 Report

Comments and Suggestions for Authors

The manuscript by Istrate-Ofiteru and co-authors is a retrospective study that evaluates a large number of endometriosis patient cases. Researchers evaluate clinical characteristics, tissue pathology, levels of inflammation, neovascularization, hormone receptivity, and oncogenic protein expression in patients with different variants of endometriosis and endometriosis-associated malignancies. The study is of great interest to the field, given that endometriosis and endometriosis-associated cancers lack clear early diagnosis indicators and require more knowledge on the basic cell and molecular mechanisms that drive the progression and malignant transformation of this group of diseases. The study is relevant to the scope of journal, however, it requires significant text and figure/chart improvement/reorganization before it can be considered for publication in IJMS.

Specific comments:

1)      It is not clear whether authors categorize Adenomyosis as a subtype of Endometriosis or consider it a separate disease. While most of the charts/figures imply Andenomyosis as one of the Endometriosis categories, in text, the two are often placed together as “Endometriosis and Adenomyosis” as if they are two separate diseases, while not mentioning other categories. It is best to clarify the Adenomyosis in the Introduction more and keep it consistent throughout the manuscript.

2)      Page 3, Chart 1 is hard to follow. It is better to transform the chart into a simple Table 1 with complete de-abbreviated nomenclature of diagnosis/pathology.

3)      Pages 4-5 and Chart 2 are very difficult to read. The chart structure is very awkward and misleading. It is obvious that some patients have multiple symptoms, but the chart is structured in a way that the different symptoms are all summarized together when it is not appropriate. E.g., 450% of patients with UE-EC-G1 is not possible. I would suggest separating the chart into figures “a”, “b”, “c”, etc. If researchers want to show differences in symptoms between histopathological groups or show the distribution of the same clinical symptoms between patients with different path reports, it’s better to separate charts by symptoms, presence of metastases, location of endometriosis, etc.

4)      Figure 1,2, 3, – how statistically relevant are these images? Researchers studied 243 cases total, and if they show “representative” images, it would be helpful to know what % of patients with one or the other sub-type of endometriosis-related condition show such gross morphology or imaging disease markers. Authors claim, “in some cases…”, “in other cases…” – what is the exact % of one or the other clinical, surgical, or paraclinical (instrumental) picture?

5)      Charts 2-8 must be visually improved!

6)      Pages 12-21 are extremely hard to follow and mostly contain numerical and abbreviated text with stats => all those can be transformed into Tables to improve the legibility and comprehension of the analysis.

7)      Figures 4 and 5 will benefit from adding arrows and arrowheads to show specific differentially stained cellular components within the H&E images with additional explanations in the Figure Legends for readers with color vision deficiency or those who have access to b/w copy.

Comments on the Quality of English Language

N/a

Author Response

Attached is the document mentioning the changes made to the manuscript. Thank you! 

Reviewer 2 Report

Comments and Suggestions for Authors

The paper is a mess. It is not evident where the Authors want to drive the reader. In the same paper they mix pain , infertility (primary and secondary), precancerous and cancerous lesions so that at the end it is very difficult to understand the purpose of the study. My suggestion is to separate the data into different paper each dealing with one specific topic  

Comments on the Quality of English Language

The quality is not the problem even if a revision should be performed

Author Response

(The authors gave the same response as above.)

Round 2

Reviewer 1 Report

Comments and Suggestions for Authors

The revised version of the manuscript by Istrate-Ofiteru and co-authors addressed the original comments. The text, and in particular, tables, histo micrographs and charts were improved and made it a bit easier for the reader to comprehend. On the other hand, there are several paragraphs that look more confusing after the additional text incorporation.

Specific comments which MUST be addressed before the paper is reconsidered:

-          In the Introduction, the authors decided to include “E and A” everywhere in text making the original statements quite incorrect. For example, revised line 74: “Pathologist can differentiate them by the presence or absence of stromal elements” or line 92 “E or A foci may present as … lesions on the ovarian surface, parietal or visceral peritoneum”. According to commonly accepted definitions and locations of Endometriosis and Adenomyosis, both of these statements are incorrect. Adenomyosis is diagnosed based on the presence of glands inside the myometrium (confirmed pathologically), it is not present on ovarian surface or peritoneum. Endometriosis (confirmed pathologically) is the growth of endometrial-like lesions outside the uterus.

-          Related to the previous comment, authors must explain what they mean by “uterine endometriosis”, given that E. implies ECTOPIC growth (not inside the uterus) and authors refer to “uterine endometriosis” and “adenomyosis” as two separate groups. Do they mean the growth of E. lesions on the serous side of the uterus? IF so, how is it different from abdominal endometriosis? It is still very confusing why the authors constantly merge E and A in the same sentences and even in the same study.

-          It would be quite beneficial to expand the newly revised Table 1 by adding a column where authors precisely indicate which E. or A. locations they consider UE-H, UE, etc. For example, lines 157-168 of the newly revised manuscript – how is “abdominal endometriosis” on line 158 is different from the “abdominal wall endometriosis” in line 165? Why is “abdominal endometriosis with hyperplastic lesions” abbreviated as UE-AH (lines 158-159) of the revised manuscript.

-          The only information about the samples obtained is provided for the “control” samples which are stated to be taken from the individuals suffering car accidents. The nature of 75% other cases (“pathological”) ones are not described at all.

-          For all the new statements made by authors in the Intro of the newly revised manuscript, they must provide references.

Minor comments:

-          Newly revised Charts 1A-E, Charts 3A-E – remove the trendlines everywhere, as they are not necessary and do not convey the point. These are bar graphs representing totally different groups and a trendline between them is not needed (this is not a correlation scatter plot, for example).

-          Have another round of English grammar check (maybe with the employment of professional resources)

Comments on the Quality of English Language

see above